# A Review on the Genus *Paramacrobiotus* (Tardigrada) with a New Diagnostic Key

Pushpalata Kayastha [1,*], Monika Mioduchowska [2], Jędrzej Warguła [1] and Łukasz Kaczmarek [1]

[1] Department of Animal Taxonomy and Ecology, Faculty of Biology, Adam Mickiewicz University, Uniwersytetu Poznańskiego 6, 61-614 Poznań, Poland; jenwar@st.amu.edu.pl (J.W.); kaczmar@amu.edu.pl (Ł.K.)

[2] Department of Evolutionary Genetics and Biosystematics, Faculty of Biology, University of Gdańsk, 59 Wita Stwosza, 80-308 Gdańsk, Poland; monika.mioduchowska@ug.edu.pl

* Correspondence: pushpalata.kayastha@gmail.com

**Abstract:** *Paramacrobiotus* species have been described in almost every corner of the world. To date, 45 species have been reported from this genus. Among which, 13 belong to the *areolatus* group (without a microplacoid) and 32 belong to the *richtersi* group (with a microplacoid). The species' presence in different climatic conditions and habitats provides evidence of their adaptation to various harsh environments. The species of the genus are both bisexual (diploid) and parthenogenetic (triploid). The bisexual species have external fertilization. And they are omnivorous whose diet consists of certain cyanobacteria, algae, fungi, rotifers, nematodes and juvenile tardigrades. The life history of species from this genus varies from species to species. Because the species has a strong predilection for cryptobiosis, numerous investigations involving anhydrobiosis have been conducted utilizing specimens from varied *Paramacrobiotus* species to date. In this review, we provide a concise summary of changes observed due to various cryptobiotic conditions in many species of this genus, the geographical distribution of all the species, feeding behaviour, life history, microbiome community, *Wolbachia* endosymbiont identification, reproduction, phylogeny and general taxonomy of the species from the genus *Paramacrobiotus*. Furthermore, we provide a new diagnostic key to the genus *Paramacrobiotus* based on the morphological and morphometric characters of adults and eggs.

**Keywords:** tardigrade; reproduction; taxonomy; distribution; microbiome





## 1. Introduction

Tardigrades, also called water bears, is a phylum consisting of ca. 1500 species [1–4] that inhabit terrestrial and aquatic environments throughout the world [5]. They are mostly found in mosses, lichens, soil, leaf litter, sediments and on aquatic plants [5–7]. The phylum consists of two classes, i.e., Heterotardigrada and Eutardigrada [5]. Eutardigrada is further divided into two limnoterrestrial orders, i.e., Apochela and Parachela. Moreover, the order Parachela consists of various superfamilies and families, one of them being Macrobiotidae (Thulin, 1928) [8] with the genus *Paramacrobiotus* Guidetti, Schill, Bertolani, Dandekar and Wolf, 2009 [9]. The genus was erected in 2009 from the genus *Macrobiotus*. These two genera are distinguished by morphological characteristics such as egg processes' shape (large and reticulated cones or trunk-cones in the genus *Paramacrobiotus*, smooth and inverted goblet shaped in the genus *Macrobiotus*). Next, only the genus *Paramacrobiotus*' buccal armature has a posterior line of strong triangular or bicuspidal teeth. Furthermore always three, well-separated macroplacoids in the *Paramacrobiotus* species are present but mostly two, and in rare cases three with the first two very close, in *Macrobiotus* species are present. Also, cuticular pores are absent in *Paramacrobiotus* but present in *Macrobiotus*. Lastly, the shape of the spermatozoa in the *Paramacrobiotus* species is such that the head is thin and very long, up to 100 μm, and it is longer than the tail; in the *Macrobiotus* species, the head is strongly coiled and long but shorter than the tail, and it has a huge

midpiece [9]. To date, 45 species have been described: *Paramacrobiotus alekseevi* (Tumanov, 2005) [10]; *Pam. arduus* Guidetti, Cesari, Bertolani, Altiero & Rebecchi, 2019 [11]; *Pam. areolatus* (Murray, 1907) [12]; *Pam. beotiae* (Durante Pasa & Maucci, 1979) [13]; *Pam. celsus* Guidetti, Cesari, Bertolani, Altiero & Rebecchi, 2019 [11]; *Pam. centesimus* (Pilato, 2000) [14]; *Pam. chieregoi* (Maucci & Durante Pasa, 1980) [15]; *Pam. corgatensis* (Pilato, Binda & Lisi, 2004) [16]; *Pam. csotiensis* (Iharos, 1966) [17]; *Pam. danielae* (Pilato, Binda, Napolitano & Moncada, 2001) [18]; *Pam. danielisae* (Pilato, Binda & Lisi, 2006) [19]; *Pam. depressus* Guidetti, Cesari, Bertolani, Altiero & Rebecchi, 2019 [11]; *Pam. derkai* (Degma, Michalczyk & Kaczmarek, 2008) [20]; *Pam. experimentalis* Kaczmarek, Mioduchowska, Poprawa & Roszkowska, 2020 [21]; *Pam. fairbanksi* Schill, Förster, Dandekar & Wolf, 2010 [22]; *Pam. filipi* Dudziak, Stec & Michalczyk 2020 [23]; *Pam. gadabouti* Kayastha, Stec, Mioduchowska and Kaczmarek 2023 [24]; *Pam. garynahi* (Kaczmarek, Michalczyk & Diduszko, 2005) [25]; *Pam. gerlachae* (Pilato, Binda & Lisi, 2004) [16]; *Pam. halei* (Bartels, Pilato, Lisi & Nelson, 2009) [26]; *Pam. hapukuensis* (Pilato, Binda & Lisi, 2006) [19]; *Pam. huziori* (Michalczyk & Kaczmarek, 2006) [27]; *Pam. intii* Kaczmarek, Cytan, Zawierucha, Diduszko & Michalczyk, 2014 [28]; *Pam. kenianus* Schill, Förster, Dandekar & Wolf, 2010 [22]; *Pam. klymenki* Pilato, Kiosya, Lisi & Sabella, 2012 [29]; *Pam. lachowskae* Stec, Roszkowska, Kaczmarek & Michalczyk, 2018 [30]; *Pam. lorenae* (Biserov, 1996) [31]; *Pam. magdalenae* (Michalczyk & Kaczmarek, 2006) [27]; *Pam. metropolitanus* Sugiura, Matsumoto & Kunieda, 2022 [32] *Pam. palaui* Schill, Förster, Dandekar & Wolf, 2010 [22]; *Pam. peteri* (Pilato, Claxton & Binda, 1989) [33]; *Pam. pius* Lisi, Binda & Pilato, 2016 [34]; *Pam. priviterae* (Binda, Pilato, Moncada & Napolitano, 2001) [35]; *Pam. richtersi* (Murray, 1911) [36]; *Pam. rioplatensis* (Claps & Rossi, 1997) [37]; *Pam. sagani* Daza, Caicedo, Lisi & Quiroga, 2017 [38]; *Pam. savai* (Binda & Pilato, 2001) [39]; *Pam. sklodowskae* (Michalczyk, Kaczmarek & Węglarska, 2006) [40]; *Pam. spatialis* Guidetti, Cesari, Bertolani, Altiero & Rebecchi, 2019 [11]; *Pam. spinosus* Kaczmarek, Gawlak, Bartels, Nelson & Roszkowska, 2017 [41]; *Pam. submorulatus* (Iharos, 1966) [17]; *Pam. tonollii* (Ramazzotti, 1956) [42]; *Pam. vanescens* (Pilato, Binda & Catanzaro, 1991) [43]; *Pam. walteri* (Biserov, 1997/98) [44]; and *Pam. wauensis* (Iharos, 1973) [45]. Furthermore, the genus is divided into two species groups, i.e., the *richtersi* group, with the presence of a microplacoid within the pharynx, and the *areolatus* group, without a microplacoid within the pharynx. In turn, Kaczmarek et al. [41] proposed separating subgenera, for which specific names were clarified by Marley et al. [46]. However, the two subgenera are not valid according to Guidetti et al. [11] and Stec et al. [47].

In this paper, we summarize the data on the taxonomy, distribution, mode of reproduction, microbiome study, feeding behaviour, life history, morphological taxonomy, phylogeny and cryptobiotic studies, along with providing a new key for species identification in the genus *Paramacrobiotus*.

## 2. Morphological Taxonomy

The genus *Paramacrobiotus* is divided into two morphologically distinct species groups: *areolatus* (species without a microplacoid or with rudimentary structures in the place of microplacoid in the pharynx) and *richtersi* (species with a microplacoid in the pharynx) (e.g., [23,28]). It was suggested that the microplacoid was initially present but was lost in some species from the *areolatus* group. But, the opposite situation, in which the microplacoid gradually appeared, is also possible [41]. For example, in *Pam. vanescens*, the microplacoid suggests a gradual reduction. In turn, in *Pam. areolatus* and *Pam. centesimus*, the microplacoid is generally absent, but a thin cuticular thickening is present in the place where the microplacoid should normally be present and can be considered as a rudimentary microplacoid [14,47]. Although the presence or absence of the microplacoid seems to be a clear morphological character dividing the genus *Paramacrobiotus* into two separate phylogenetic lineages (which was suggested by Kaczmarek et al. [41]), but genetic studies did not confirm this [11,47].

At present, 45 species are formally attributed to the genus *Paramacrobiotus*, 13 belong to the *areolatus* group, and 32 belong to the *richtersi* group. They can be further divided into

smaller groups based on egg types. In total, seven types of eggs were identified. However, two of them (*areolatus* and *richtersi* types) are the most common and occur in 37 species (*ca.* 82%). In the next two species, the *huziori* type of eggs are present (ca. 5%). The other types of eggs (i.e., *beotiae, chieregoi, csotiensis, tonollii* and *submorulatus*) were identified only in single taxa (for details of egg morphology, see Kaczmarek et al. [41]). Furthermore, eggs are unknown for one species, *Pam. wauensis*.

In recent years, two very important species for taxonomy of the entire genus, *Pam. areolatus* and *Pam. richtersi*, were integratively redescribed [11,47]. Another species, *Pam. fairbanksi*, described based mostly on genetic data, was also morphometrically well characterized a few years ago [21]. However, a few *Paramacrobiotus* species still need a redescription based on the type material or on additional material from type localities. Descriptions of *Pam. beotiae, Pam. chieregoi, Pam. csotiensis, Pam. rioplatensis, Pam. submorulatus, Pam. tonollii* and *Pam. wauensis* are inaccurate, and some important morphological informations are lacking.

Another two species, i.e., *Pam. kenianus* and *Pam. palaui*, are cryptic taxa described mostly based on genetic data without morphological differential diagnosis [22].

Descriptions of the other *Paramacrobiotus* species are more or less complete, but in most of them, exact morphometric data of claws and buccal tubes placoids and, above all, genetic data are lacking (see Table 1 and Supplementary Materials SM.01). Based on all the abovementioned doubts, three species, i.e., *Pam. kenianus, Pam. palaui* and *Pam. wauensis*, are not included in the key.

**Table 1.** Selected morphological characters of the known species of genus *Paramacrobiotus* (schematic illustrations of different types of egg process shapes presented in Figure 1).

| Species | Cuticle | Number of Rows in Oral Cavity Armature | Eyes | Lunules IV | Granulation on Legs | Egg Type | Egg Process Height (in μm) | Egg Process Base Width (in μm) | Egg Process Shape | Number of Processes on Circumference |
|---|---|---|---|---|---|---|---|---|---|---|
| *Paramacrobiotus alekseevi* | smooth | I–III | absent | dentate | IV | *richtersi* | 11.8–21.8 | 13.3–22.9 | cone with cap | 10–12 |
| *Paramacrobiotus arduus* | smooth | I–III | absent | smooth | I–IV | *richtersi* | 12.1–18.3 | 10.4–16.3 | simple cone | 16–21 |
| *Paramacrobiotus areolatus* | smooth | I–III | present | crenate | I–IV | *areolatus* | 20.0–28.0 | 19.0–22.0 | simple cone | ? |
| *Paramacrobiotus beotiae* | smooth | I–III | absent | dentate | ? | *beotiae* | up to 16.0 | ? | spines | ? |
| *Paramacrobiotus celsus* | smooth | I–III | absent | smooth | I–IV | *richtersi* | 15.2–19.1 | 14.3–18.2 | simple cone (jagged) | 15–19 |
| *Paramacrobiotus centesimus* | smooth | I–III | absent | smooth | I–IV | *areolatus* | 7.0–11.0 | ? | simple cone | 11–12 |
| *Paramacrobiotus chieregoi* | smooth | I–III | absent | smooth | ? | *chieregoi* | ? | ? | elongated cone | 14 |
| *Paramacrobiotus corgatensis* | sculptured | I–III | present | dentate | ? | *richtersi* | 20.0–25.0 | 18.0–24.0 | simple cone (jagged) | 8–11 |
| *Paramacrobiotus csotiensis* | smooth | II–III | present | ? | ? | *csotiensis* | ? | ? | hemispherical covered with a hyaline layer | ? |
| *Paramacrobiotus danielae* | sculptured | I–III | present | smooth | ? | *areolatus* | 14.5 | 24.7 | simple cone | ? |
| *Paramacrobiotus danielisae* | sculptured | I–III | absent | smooth | ? | *richtersi* | 17.3–23.0 | 17.5–20.0 | simple cone | 9–10 |
| *Paramacrobiotus depressus* | smooth | I–III | absent | smooth | IV | *richtersi* | 9.3–12.4 | 12.4–15.2 | simple cone | 16–23 |
| *Paramacrobiotus derkai* | smooth | I–III | present | smooth | I–IV | *huziori* | 8.0–17.1 | 12.5–28.3 | simple cone | 12–16 |
| *Paramacrobiotus experimentalis* | smooth | I–III | absent | smooth | IV | *areolatus* | 10.3–14.9 | 13.8–19.4 | simple cone | 10–12 |
| *Paramacrobiotus fairbanksi* | smooth | I–III | absent | smooth | I–IV | *richtersi* | 10.9–14.9 | 10.9–20.8 | simple cone (jagged) | ? |
| *Paramacrobiotus filipi* | granulation | I–III | absent | smooth | I–IV | *richtersi* | 17.8–25.2 | 11.7–21.7 | cone with cap | 10–11 |

**Table 1.** *Cont.*

| Species | Cuticle | Number of Rows in Oral Cavity Armature | Eyes | Lunules IV | Granulation on Legs | Egg Type | Egg Process Height (in μm) | Egg Process Base Width (in μm) | Egg Process Shape | Number of Processes on Circumference |
|---|---|---|---|---|---|---|---|---|---|---|
| *Paramacrobiotus gadabouti* | smooth | I–III | absent | smooth | IV | *richtersi* | 12.1–23.7 | 15.0–25.5 | truncated cones | 11–13 |
| *Paramacrobiotus garynahi* | with pores | I–III | absent | smooth | I–IV | *areolatus* | 18.0–30.0 | 20.0–42.0 | cone with cap | 10–13 |
| *Paramacrobiotus gerlachae* | smooth | I–III | absent | smooth | IV | *richtersi* | 11.8–14.5 | 16.8–18.7 | simple cone | ? |
| *Paramacrobiotus halei* | sculptured | I–III | absent | ? | I–IV | *richtersi* | 11.0–14.0 | 22.0–23.5 | blunt cone | 11 |
| *Paramacrobiotus hapukuensis* | smooth | I–III | absent | smooth | absent | –*areolatus* | 15.7–21.1 | 14.8–16.6 | elongated cone | 10 |
| *Paramacrobiotus huziori* | smooth | I–III | present | smooth | I–IV | *huziori* | 20.0–33.0 | 20.0–30.0 | simple cone | 9–11 |
| *Paramacrobiotus intii* | smooth | II–III | present | dentate | I–IV | *areolatus* | 15.4–24.4 | 22.0–34.0 | simple cone | 9–10 |
| *Paramacrobiotus kenianus* | smooth | ? | present | ? | ? | *richtersi* | 13.5 ± 1.9 | 19.7 ± 2.7 | simple cone | 17.7 ± 3.6 |
| *Paramacrobiotus klymenki* | smooth | I–III | absent | dentate | I–IV | *areolatus* | 14.5–18.5 | 16.4–18.2 | simple cone | 10–11 |
| *Paramacrobiotus lachowskae* | smooth | I–III | present | smooth | I–IV | *areolatus* | 17.6–32.1 | 8.1–17.7 | hemispherical with filaments | 8–14 |
| *Paramacrobiotus lorenae* | smooth | I–III | absent | smooth | I–IV | *richtersi* | 25.0–42.2 | 17.8–23.0 | elongated cone | ? |
| *Paramacrobiotus magdalenae* | smooth | I–III | present | smooth | IV | *richtersi* | 13.0–25.0 | 16.2–21.0 | simple cone | 10–12 |
| *Paramacrobiotus metropolitanus* | smooth | I–III | absent | smooth | IV | *areolatus* | 7.4–14.6 | 9.8–21.1 | simple cone | 10–15 |
| *Paramacrobiotus palaui* | smooth | ? | present | ? | ? | *richtersi* | 10.2 ± 1.3 | 13.4 ± 1.3 | simple cone | 15.4 ± 1.4 |
| *Paramacrobiotus peteri* | smooth | I–III | absent | smooth | ? | *areolatus* | 10.0–14.0 | 9.0–12.0 | simple cone (jagged) | ? |
| *Paramacrobiotus pius* | smooth | I–III | absent | smooth | I–IV | *richtersi* | up to 12.3 | 19.5–24.7 | simple cone (jagged) | 10 |
| *Paramacrobiotus priviterae* | smooth | I–III | present | smooth | I–IV | *richtersi* | 11.8–15.0 | 12.9–16.3 | simple cone (jagged) | ? |

**Table 1.** *Cont.*

| Species | Cuticle | Number of Rows in Oral Cavity Armature | Eyes | Lunules IV | Granulation on Legs | Egg Type | Egg Process Height (in μm) | Egg Process Base Width (in μm) | Egg Process Shape | Number of Processes on Circumference |
|---|---|---|---|---|---|---|---|---|---|---|
| *Paramacrobiotus richtersi* | smooth | I–III | absent | smooth | I–IV | *richtersi* | 17.1–22.1 | 17.2–22.2 | simple cone | 13–17 |
| *Paramacrobiotus rioplatensis* | smooth | I–III | present | smooth | ? | *areolatus* | ca. 4.6 | ? | elongated cone | 17–19 |
| *Paramacrobiotus sagani* | granulation | I–III | present | smooth | I–IV | *richtersi* | 9.4–13.2 | 14.6–22.4 | blunt cone | 10–13 |
| *Paramacrobiotus savai* | smooth | I–III | present | smooth | IV | *areolatus* | 12.0–18.0 | 16.7–18.5 | blunt cone | ? |
| *Paramacrobiotus sklodowskae* | smooth | I–III | present | smooth | I–IV | *richtersi* | 16.0–17.5 | 20.5–23.5 | blunt cone | 10 |
| *Paramacrobiotus spatialis* | smooth | I–III | absent | smooth | I–IV | *richtersi* | 13–16 | 15.2–20.4 | simple cone | 15–23 |
| *Paramacrobiotus spinosus* | smooth | I–III | absent | smooth | I–IV | *richtersi* | 22.1–42.2 | 17.3–26.0 | elongated cone (jagged) | 10–11 |
| *Paramacrobiotus submorulatus* | smooth | II–III | present | ? | ? | *submorulatus* | 7.0–8.3 | 17.5–20.4 | hemispherical with concave on top | 13 |
| *Paramacrobiotus tonollii* | smooth | ? | present | smooth | ? | *tonollii* | 32.0–35.0 | ? | elongated cone | 8–10 |
| *Paramacrobiotus vanescens* | faint punctuation | I–III | absent | ? | I–IV | *richtersi* | 16.0–17.0 | 24.0–25.0 | blunt cone (jagged) | 9–12 |
| *Paramacrobiotus walteri* | smooth | I–III | present | dentate | I–IV | *areolatus* | 10.0–17.0 | 9.0–20.0 | simple cone (jagged) | ? |
| *Paramacrobiotus wauensis* | smooth | I–III | absent | ? | ? | ? | ? | ? | ? | ? |

Note: I–IV represents the number of pair of legs and ? means unsuitable or not present.

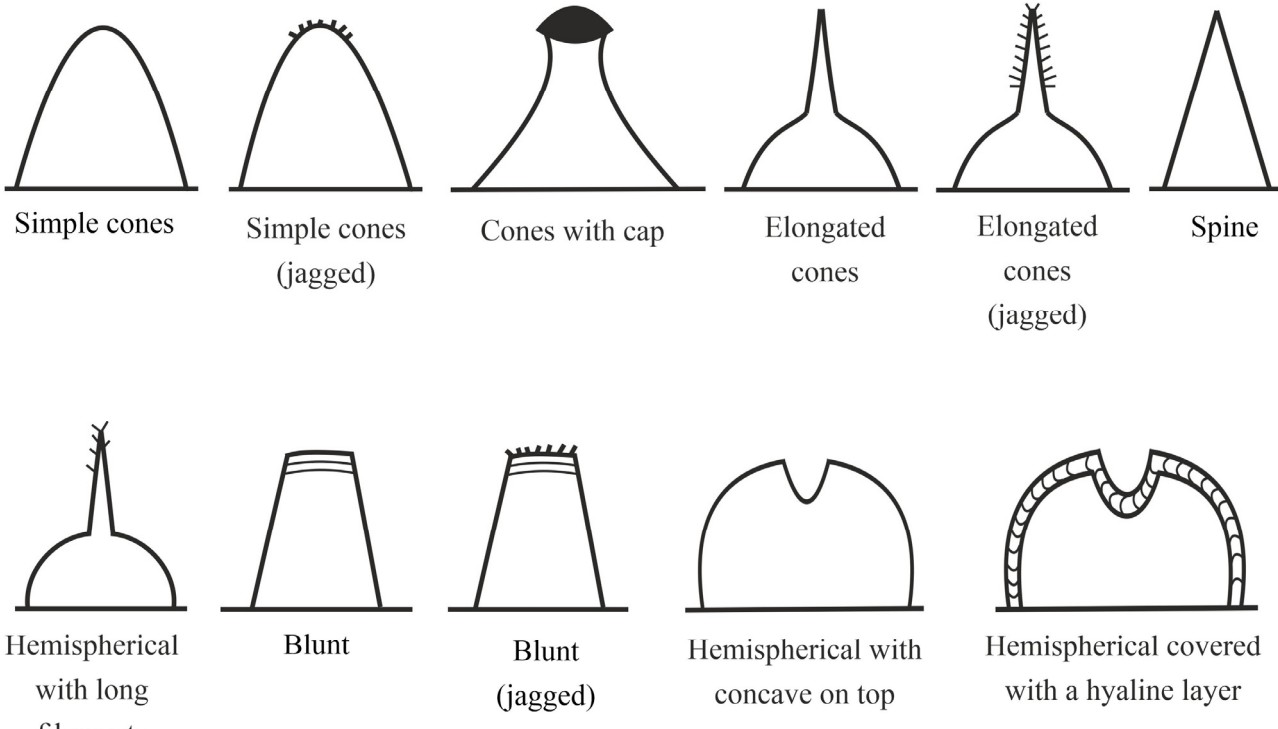

**Figure 1.** Schematic illustrations of the types of egg processes in the genus *Paramacrobiotus*.

### 3. Molecular Taxonomy

Molecular markers serve as valuable tools for species identification. In the integrative taxonomy of Tardigrada, four DNA fragments with different mutation rates are commonly used: two conservative nuclear ribosomal subunit genes, namely 18S rRNA (the small ribosome subunit) and 28S rRNA (the large ribosome subunit); the noncoding nuclear ITS-2 fragment (the internal transcribed spacer-2) with high evolution rates; and the protein-coding mitochondrial COI barcode gene (the cytochrome oxidase subunit I), with an intermediate effective mutation rate (e.g., [48]). The COI mtDNA molecular marker, in particular, has been recommended for DNA barcoding purposes (http://www.barcodinglife.org accessed on 10 July 2023), such as rapid species identification, discrimination between cryptic species, and resolving phylogenetic relationships among closely related species [49,50]. To gain additional insights into the phylogenetic relationships within the genus *Paramacrobiotus*, an analysis based on COI mtDNA was conducted. This analysis was performed to supplement the information obtained from previous studies using four molecular markers [24].

Due to ongoing revisions and redescriptions of *Paramacrobiotus* species, studies are becoming more accessible, leading us to anticipate that the species diversity within the genus is greatly underestimated [11,23]. One significant challenge that needs to be addressed in future studies is the lack of available barcodes. Despite the designation of 45 species in the genus *Paramacrobiotus*, not all species have available barcode sequences. In this study, we aimed to estimate the phylogenetic relationships among all *Paramacrobiotus* species (including taxa designated as "cf.", meaning "compare with", and "aff.", meaning "similar to") for which COI barcode sequences are available in the GenBank database. The alignment of COI barcode sequences resulted in 574 characters, with 270 variable sites and 241 parsimony informative sites. We used the COI sequence of *Milnesium berladnicorum* Ciobanu, Zawierucha, Moglan & Kaczmarek, 2014 [51] as the outgroup to construct the most reliable evolutionary tree. To determine the most appropriate model of sequence evolution, we applied jModelTest v. 2.1.4 [52] with both the Bayesian Information Criterion (BIC) and the Akaike Information Criterion (AIC) [53]. The GTR + G (Time-Reversible model with gamma-distributed rate heterogeneity) was selected as the best-fitting evolutionary model.

The phylogenetic tree was constructed using (i) Bayesian inference (BI) analysis with the program MrBayes 3 [54], following the settings described by Mioduchowska et al. [55], and (ii) maximum likelihood (ML) analysis calculated using the program Mega X [56] with 1000 bootstraps and under the general settings of the selected evolutionary model. Uncorrected pairwise distances (p-distances) were calculated using MEGA X [56].

The binary model of phylogenetic relationships, which involves reconstructing gene trees from sequence data, allows us to gain insights into the speciation history of species [57]. However, in our analysis of barcode sequences, we observed speciation events that resulted in polytomies within the phylogeny of the genus *Paramacrobiotus* (Figure 2). This means that more than two descendants were observed from certain nodes [58]. The presence of unresolved nodes in a polytomic multifurcating tree indicates a lack of a signal in the data to resolve relationships within the genus *Paramacrobiotus*. This observation is partially consistent with previous studies, where both groups, *richtersi* and *areolatus*, were described as polyphyletic [11,47]. However, in the work by Kayastha et al. [24], the interrelationships of the genus *Paramacrobiotus* were not depicted as a polytomy when two conservative coding nuclear molecular markers (18S rRNA and 28S rRNA) and a noncoding nuclear marker with high evolution rates (ITS2) were included in the analysis. As a result, the phylogenetic relationships within the genus *Paramacrobiotus* were resolved. Interestingly, other examples of polytomies in Tardigrada gene trees based on nuclear molecular markers have also been observed [59]. In turn, Stec et al. [47] performed a cross-strain experiment to observe the molecular taxonomy of the genus *Paramacrobiotus* and indicated hidden species richness. The authors concluded that the utilization of DNA barcodes may prove inadequate in fully resolving species diversity and accurately describing species within this cosmopolitan genus. Hence, both multilocus sequencing and direct experimental testing of species boundaries are required.

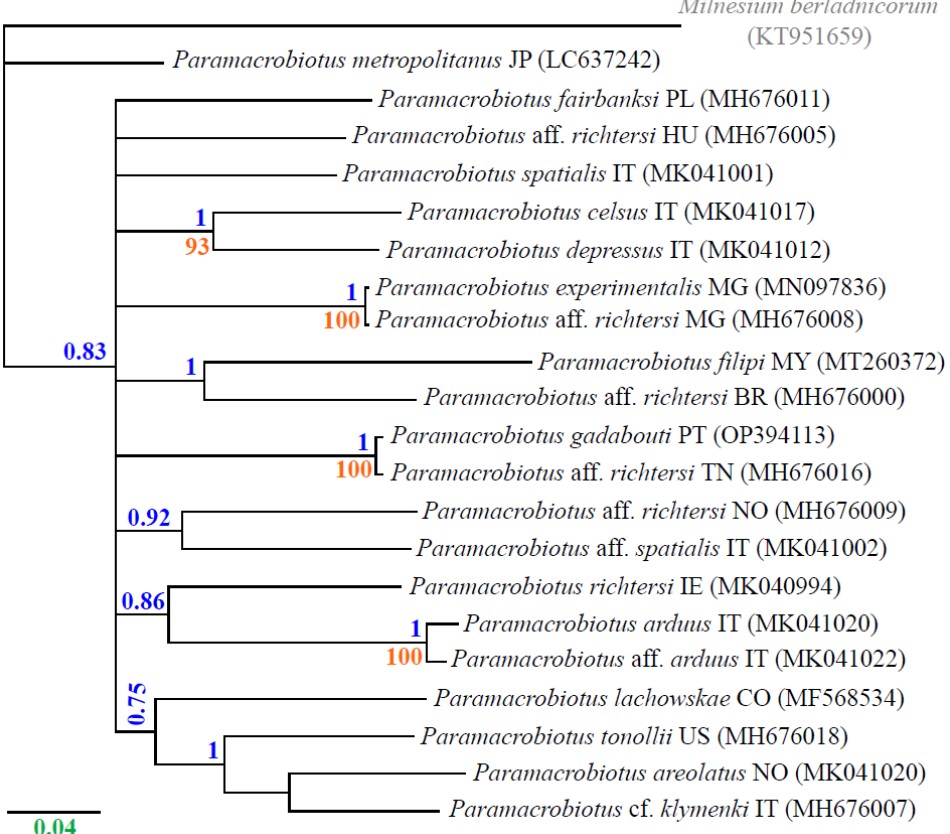

**Figure 2.** Phylogenetic relationships of the genus *Paramacrobiotus* constructed based on the COI barcode sequences obtained from the GenBank database. The GenBank accession numbers are given

in parentheses. In turn, locations of identified species are given in abbreviations: JP—Japan; PL—Poland; HU—Hungary; IT—Italy; MG—Madagascar; MY—Malaysia; BR—Brazil; PT—Portugal; TN—Tunisia; NO—Norway; IE—Ireland; CO—Colombia; US—United States. The numbers above the branches represent Bayesian posterior probabilities, and the supporting bootstrap values from the maximum likelihood analysis are provided beneath the branches. Branches with support below 70% in ML and below 0.7 in BI were collapsed. The COI sequence of *Milnesium berladnicorum* was used as an outgroup.

The genetic p-distances between the analyzed COI barcode sequences of *Paramacrobiotus* species ranged from 16% to 27%, indicating different species (Supplementary Materials SM.02). However, it was shown that there are very low genetic differences, i.e., a p-distance of 0.3%, between *Pam*. aff. *richtersi* from Tunisia (GenBank: MH676016) and *Pam. gadabouti* from Portugal (GenBank: OP394113), suggesting they belong to the same species (Supplementary Materials SM.02). This finding is consistent with the work by Kayastha et al. [24], where both species were described as *Pam. gadabouti*. No genetic differences were found between *Pam*. aff. *richtersi* from Madagascar (GenBank: MH676008) and *Pam. experimentalis* from Madagascar (GenBank: MN097836) (Supplementary Materials SM.02). Both sequences represented *Pam. experimentalis*, which is also consistent with the previous study [24]. Moreover, we found very low genetic differences, i.e., a p-distance of 2.1%, between *Pam. arduus* from Italy (GenBank: MK041020) and *Pam*. aff. *arduus* from Italy (GenBank: MK041022), indicating the same species (Supplementary Materials SM.02).

## 4. Cryptobiosis

The stage of an organism's life known as cryptobiosis is one in which no activity is apparent [60]. Many organisms go through cryptobiosis to survive the harsh environmental conditions they encounter [61–63]. A few types of cryptobiosis are known i.e., anhydrobiosis (lack of water), anoxybiosis (lack of oxygen), cryobiosis (low temperature), or osmobiosis (change in osmotic conditions). Tardigrades have a remarkable capacity for undergoing and surviving several types of cryptobiosis [60,64]. In genus *Paramacrobiotus*, majority of studies related to cryptobiosis are anhydrobiosis, or the absence of water, additionally, there has also been research on famine, freezing, and bet-hedging [65–69]. Reuner et al. [65] studied how the influence of starvation and anhydrobiosis affects the size and number of storage cells in *Pam. tonollii* to understand the energetic side of anhydrobiosis. Starving *Pam. tonollii* for seven days led to a reduction in storage cell size by 46.41%, but no significant reduction in storage cell number was observed. Furthermore, when storage cells' size and number were investigated after inducing anhydrobiosis for seven days, no significant changes in storage cell size or its number in *Pam. tonollii* were observed. Also, the mortality was checked using prolonged starvation, and *Pam. tonollii* reached 50% mortality after 30 days. Likewise, Rizzo et al. [66] investigated antioxidant defences (capable of counteracting reactive oxygen species (ROS)) in *Pam. richtersi* in both active and dehydrated states. The activity of several antioxidant enzymes, the fatty acid composition, and heat shock protein (Hsp) expression were compared in these two states. The increase in both antioxidant enzymes (superoxide dismutase due to induction of both glutathione peroxidase and glutathione during desiccation) and the fatty acid composition (polyunsaturated fatty acids and the amount of substances reactive to thiobarbituric acid) were observed in desiccated *Pam. richtersi* specimens, but no significant differences in the relative level of heat shock proteins were observed (Hsp70 and Hsp90). In addition, Giovannini et al. [68] performed a study in which the production of reactive oxygen species and the involvement of bioprotectants during anhydrobiosis in *Pam. spatialis* was investigated. The study provides evidence of an increase in ROS production relative to the time spent in anhydrobiosis, which is due to oxidative stress in the animals. Using RNA interference, the involvement of bioprotectants, including those combating ROS, was assessed. As Rizzo et al. [66] concluded, the role of glutathione peroxidase in desiccation in *Pam. richtersi*, this gene was targeted, and what was observed was that glutathione peroxidase gene compromised survival during the drying

and rehydration of *Pam. spatialis*. This further strengthened the evidence that glutathione reductase and catalase play important roles during desiccation and rehydration. Also, the involvement of aquaporins 3 and 10 during the rehydration of *Pam. spatialis* was observed. And recently, Roszkowska et al. [69] studied the length of time that different tardigrades survive in the anhydrobiotic state, including *Pam. experimentalis*. The study concludes that anhydrobiotic competence is dependent on habitat instead of nutritional behaviour and the time taken to return to activity after anhydrobiosis is dependent upon the length of the anhydrobiosis. It is worth noting that in 2021, the entire genome of *Paramacrobiotus* sp., later described as *Pam. metropolitanus*, was sequenced [70]. This provides an opportunity for a better understanding of the genetic basis that enables them to survive the process of anhydrobiosis. The full DNA sequence has allowed for clues regarding the phylogeny of TPS-TPP genes responsible for the production of trehalose, a substance involved in anhydrobiosis. Four years earlier, in 2017, a similar mechanism was described in the action of the TDP protein [71]. Research conducted on the species *Pam. richtersi* [9], among others, revealed the involvement of this compound in DNA protection during the gradual dehydration of the organism. Further studies on the genomes of tardigrades from the genus *Paramacrobiotus* have the potential to uncover new information that can contribute not only to understanding the unique characteristics of these organisms, but also to gaining broader insights into evolution and the adaptive plasticity of organisms in various extreme environments.

## 5. Distribution

*Paramacrobiotus* species shows worldwide distribution. However, the real distribution of the *Paramacrobiotus* species is unknown due to taxonomic problems, misidentifications and lack of genetic data. This is especially visible for often reported species like *Pam. areolatus* or *Pam. richtersi*. Most of the reports of these two taxa belong to a different species. Here we present a confirmed distribution (reports from type localities or with genetic confirmation) of all 45 species in the genus *Paramacrobiotus* to date (in Supplementary Materials SM.01 and Figure 3).

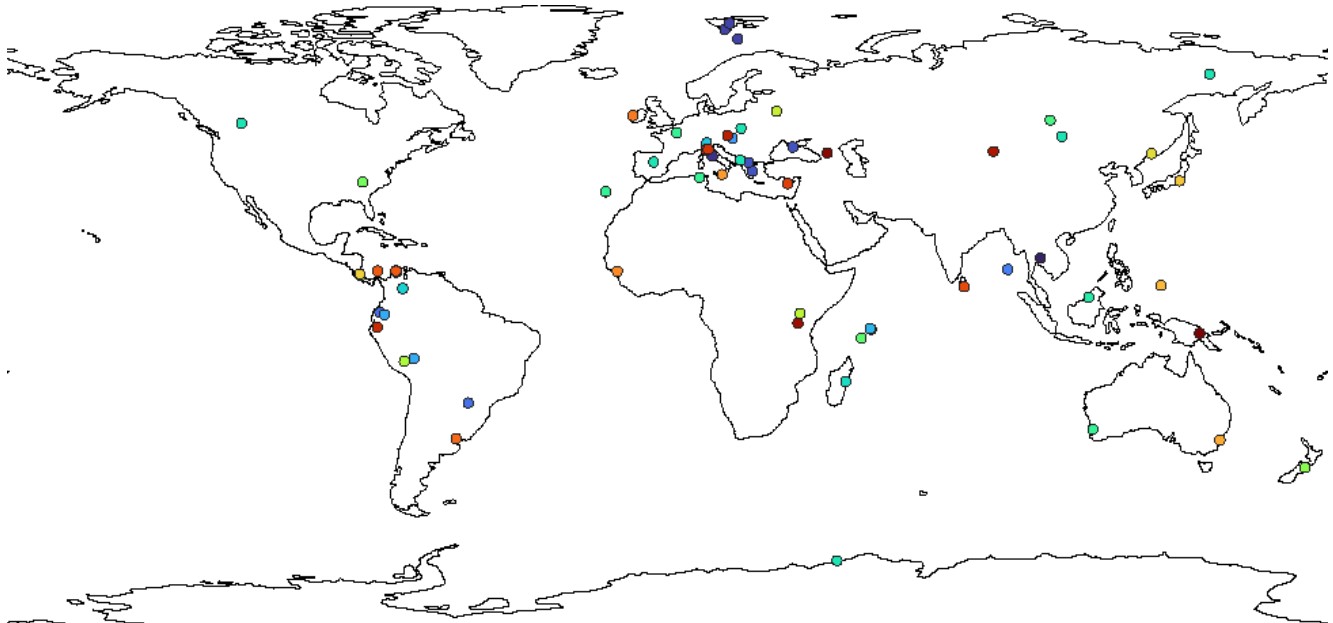

**Figure 3.** Distribution of all the species in the genus *Paramacrobiotus* (coordinates and color legend are presented in the Supplementary Materials SM.01). (Map prepared using QGIS ver. 3.28.0-Firenze.)

## 6. Feeding Behaviour

*Paramacrobiotus* species are omnivorous and consume a variety of organisms, including certain cyanobacteria, algae, and fungi, as well as the rotifer, nematodes, and small juvenile tardigrades. Additionally, the diets of adults and juveniles differ: adults favour rotifers and nematodes, whereas juveniles favour unicellular green algae. Moreover, juveniles suck out all of them, including algal cells, animal food, and fungal cells, in contrast to adults, who only consume entire fungal and algal cells [72].

## 7. Life History

Life history refers to total lifespan, development, reproduction and death of an organism [73]. The life history list in case of tardigrades consists of age at first oviposition, clutch size, fecundity, hatching percentage, hatching success, lifespan, moulting number and total number of ovipositions [74,75]. The lifespan differs from species to species in the case of tardigrades [76]. The life histories of only a few *Paramacrobiotus* species have been reported to date, that is, *Pam. fairbanski*, with an average lifespan of $137.3 \pm 136.4$ days and $194.9 \pm 164.4$ days and age at first oviposition of $70.7 \pm 19.4$ days and $76.9 \pm 16.4$ days [77]; *Pam. kenianus*, with an average lifespan of $125 \pm 35$ days and $141 \pm 54$ days, a maximum lifespan of 204 days and 212 days, and age at first oviposition of 10 days and 10 days [74]; *Pam. metropolitanus*, with juveniles hatching in 12–20 days and first oviposition within 11–13 days after hatching [78]; *Pam. palaui*, with an average lifespan of $97 \pm 31$ days, a maximum lifespan of 187 days, and age at first oviposition of 10 days [60]; *Pam. richtersi*, with an age at first oviposition of $64.2 \pm 1.7$ days [79]; and *Pam. tonollii*, with an average lifespan of $69.0 \pm 45.1$ days, a maximum lifespan of 237 days, and an age at first oviposition of $24.4 \pm 4.4$ days [76].

## 8. Microbiome

The microbiome represents the entire community of microorganisms, including fungi, protists, bacteria, archaea, as well as that inhabit all known metazoan species. The bacterial component of the microbiome community plays crucial roles in multiple aspects of ecdysozoan host life, such as behaviour, metabolism, development, immunity, or pathogen defence, thereby regulating the functioning of the entire organism [80,81]. Conversely, it has also been demonstrated that the host's phylogeny [82] and diet [83] have significant impacts on the overall microbial composition. Indeed, many metazoan species appear to harbour their own specific microbiome community [48]. However, our understanding of the microbiome composition of Tardigrada, based on next-generation sequencing (NGS) methods targeting the standard 16S rRNA bacterial barcoding gene fragment, is limited to a very small number of published articles [84–90].

In the case of species from the genus *Paramacrobiotus*, the microbiomes of a few species have been studied to date. In 2018, Vecchi et al. [84] described the bacterial communities associated with six limno-terrestrial tardigrade taxa, one of which was *Pam. areolatus*. The study revealed that the microbial community was mainly composed of Proteobacteria and Bacteroidetes. Interestingly, certain classified Operational Taxonomic Units (OTUs) showed variations among species from geographically distant samples. However, in all the investigated species' microbiome profiles, the order Rickettsiales was consistently identified. This order belongs to the class Alphaproteobacteria and is characterized by both pathogens and intracellular mutualists [91]. There were two distinct patterns in the diversity observed between tardigrades and their substrates, indicating significantly less microbial diversity in tardigrades compared to their substrates. This phenomenon may be attributed to tardigrades selectively associating with specific microbial communities that promote the growth of certain bacterial species while inhibiting others. Another hypothesis suggests that substrates, being complex matrices with wide surface areas and volumes, can support high bacterial biomass, resulting in a vast and complex microbial community.

Similarly, Kaczmarek et al. [21] conducted a microbiome analysis on two populations of *Pam. experimentalis* from Madagascar and their laboratory culture environment. These populations of *Pam. experimentalis* had been maintained in laboratory culture for two years. The most abundant phylum in all samples was Proteobacteria. Firmicutes was the second most dominant phylum in both *Pam. experimentalis* populations, while Bacteroides was the second most dominant phylum in the laboratory habitat. With the exception of the phyla Verrucomicrobia and Saccharibacteria, which were not found in the tardigrade microbiome, all identified taxa in the *Pam. experimentalis* microbiome community and laboratory culture environment were widespread and had comparable abundances. This confirms that the tardigrade microbiome significantly differs in composition from the bacteria inhabiting their environment. Moreover, within the microbiome of *Pam. experimentalis*, OTUs classified as potential endosymbionts belonging to the order Rickettsiales were identified. The absence of Rickettsiales OTUs in the environment of the studied species further supports the close association of these bacteria with their host.

Furthermore, Mioduchowska et al. [88] conducted a study to investigate whether tardigrade species are infected with bacterial endosymbionts belonging to the genus *Wolbachia*. The analysis included *Pam. fairbanksi* and *Paramacrobiotus* sp. In the study, Proteobacteria, Firmicutes, and Actinobacteria were identified as the three most prevalent phyla among the analyzed tardigrades, including species outside the genus *Paramacrobiotus*. However, the focus of the study was on potential tardigrade endosymbionts, particularly OTUs from the order Rickettsiales and the genus *Wolbachia*. Both Rickettsiales and *Wolbachia* were detected in the adult *Paramacrobiotus* sp., while only Rickettsiales were found in *Pam. fairbanksi* eggs. Adult *Pam. fairbanksi* did not have either *Wolbachia* or Rickettsiales infections. The genus *Wolbachia* is an intracellular bacterium belonging to the order Rickettsiales, and it infects various invertebrates, particularly terrestrial insects [92]. However, recent studies have identified infections of this bacterial endosymbiont in various freshwater invertebrate species [90,93,94]. Generally, this bacterium is transmitted vertically from mother to offspring and/or through horizontal transfer directly from the environment or between different hosts [95]. Subsequently, *Wolbachia* manipulates host reproduction by inducing parthenogenesis, feminization, male killing, or cytoplasmic incompatibility [96,97].

In 2023, Mioduchowska et al. [90] described new molecular and bioinformatic tools for detecting *Wolbachia* in freshwater invertebrates. In this study, *Wolbachia* was detected in *Pam. experimentalis*, which were the same isolates analyzed by Kaczmarek et al. [85]. Phylogenetic analysis of the obtained bacterial sequences allowed for their classification within the differentiated supergroup A of the genus *Wolbachia*. The discovery of *Wolbachia* in tardigrades opens new frontiers in understanding the *Wolbachia*-driven biology and ecology of Tardigrada.

## 9. Reproduction

Reproduction refers to the process whereby every known organism produces offspring either sexually or asexually. In the case of tardigrades, they reproduce only through gametes via many different patterns, i.e., dioecious (separate male and female), hermaphroditic (single animal with both male and female reproductive parts), or parthenogenetic (a form of asexual reproduction when only females are present in the population) [98]. The genus *Paramacrobiotus* consists of both bisexual and unisexual species/populations. The *Pam. arduus* from Italy is bisexual; the *Pam. areolatus* population from Italy is bisexual; the population from Svalbard is unisexual; *Pam. celsus* from Italy is bisexual; *Pam. depressus* from Italy is bisexual; *Pam. experimentalis* from Madagascar is bisexual; *Pam. fairbanksi* from various locations such as the Antarctic, Italy, Poland, Spain and USA is unisexual; *Pam. filipi* from Borneo is unisexual; *Pam. gadabouti* from various locations in Portugal, Australia, France and Tunisia is unisexual; *Pam. kenianus* from Kenya is unisexual; *Pam. metropolitanus* from Japan is bisexual; *Pam. palaui* from Micronesia is unisexual; *Pam. richtersi* from Ireland is bisexual, and according, to modern taxonomy, probably constitutes a distinct species;

*Pam. spatialis* from Italy is bisexual; and *Pam. tonolli* from the USA is bisexual. Out of 45, the mode of reproduction for only 14 species is known (Supplementary Materials SM.01).

An important aspect of reproduction is the morphology of sperm, the types of fertilization and reproductive strategies. In bisexual *Paramacrobiotus* species, external fertilization has been observed, which occurs after the female lays eggs [9]. Sperm in this group of tardigrades are characterized by a longer acrosome compared to genera like *Mesobiotus*, *Xerobiotus* or *Macrobiotus* [32]. A similar situation occurs in the case of the tail. The size in the genus *Paramacrobiotus* ranges from 13 µm to 29.4 µm, which is considerably longer than in the genus *Macrobiotus* (9.4–24.2 µm) [99]. Such dimensions are crucial when discussing the speed of movement of male gametes, which increases with tail length [100]. However, the length of the tail can change when the sperm enters the spermatheca. In the species *Paramacrobiotus sp.* and *Macrobiotus shonaicus* (Stec, Arakwa & Michalczyk 2018) [101], such changes were observed for the first time, characterized by a shortening of the tail (1.3–3.6 µm). This reduction is natural, as once the sperm reaches the spermatheca, the tail ceases to serve its purpose, and its length becomes nonessential [99]. Within species, there are often many differences in sperm morphology (length of the nucleus, acrosome and tail), which can be potentially useful in the context of research on the taxonomy of the genus *Paramacrobiotus* [32].

Among the species in this genus, a significant correlation between reproductive strategy and karyotype has been observed [9]. For example, in different populations (from Ireland and Italy) of *Pam. richtersi*, different chromosomal compositions within the COX1 gene were found. It was observed that in the population consisting of only females (an apomictic phenomenon), animals were triploid, and they underwent ameiotic oocyte maturation. In the case of the bisexual species, individuals were diploid, with chromosomal pairing occurring during oocyte and spermatocyte maturation [9]. These observed reproductive differences, genetic studies, and variations in egg morphology allowed the distinction of four new taxa within the *Pam. richtersi* species complex.

Also, Guidetti et al. [11] suggest the mode of reproduction being related to a constrained or wide distribution of the species. The amphimictic species display a very constrained or punctiform distribution, in contrast to the parthenogenetic species' extremely extensive spread and presence over multiple continents. The difference in the ability for dispersal linked to the two modes of reproduction can be used to explain why apomictic and amphimictic populations are distributed differently.

## 10. Key for Species Identification

1. Microplacoid present (*richtersi* group) . . . . . . . . . . . . . . . . . . . . . . . . . . . . . . . . . . . .. .. . **2**

–. Microplacoid absent (*areolatus* group) . . . . . . . . . . . . . . . . . . . . . . . . . . . . . . . . . . . .. **32**

2. Cuticular pattern on the dorsal side of the body present and visible in LM (PCM and/or DIC) . . . . . . . . . . . . . . . . . . . . . . . . . . . . . . . . . . . . . . . . . . . . . . . . . . . . . . . . . . . . . . **3**

–. Cuticle on the dorsal side of the body smooth or cuticular pattern not visible in LM (PCM and/or DIC) . . . . . . . . . . . . . . . . . . . . . . . . . . . . . . . . . . . . . . . . . . . . . . . . . . . . . . . . . . . . . . **7**

3. Eggs of *areolatus* type. . . . . . . . . . . . . . . . . . . . . . . . . . . . . . . . . . . . . . . .. .. *Pam. danielae*

–. Eggs of *richtersi* type . . . . . . . . . . . . . . . . . . . . . . . . . . . . . . . . . . . . . . . . . .. . . . . . . . . . . . **4**

4. Eyes present, lunules under claws IV dentate . . . . . . . . . . . . . . . . . . . . . . . . *Pam. corgatensis*

–. Eyes absent, lunules under claws IV smooth . . . . . . . . . . . . . . . . . . . . . . . . . . . . . . . . .. **5**

5. Dorsal cuticle covered with very small circular or elongated tubercles, egg processes less than 14.5 µm height . . . . . . . . . . . . . . . . . . . . . . . . . . . . . . . . . . . . . . . . . . . . .. *Pam. halei*

–. Dorsal cuticle covered with small dots (granules) or small polygons, egg processes more than 15.5 µm height . . . . . . . . . . . . . . . . . . . . . . . . . . . . . . . . . . . . . . . . . . . . . . . . . . . **6**

6. Dorsal cuticle covered with small dots (granules) . . . . . . . . . . . . . . . . . . . . *Pam. vanescens*

–. Dorsal cuticle covered with small polygons . . . . . . . . . . . . . . . . . . . . . . . . *Pam. danielisae*

7. Areolation between egg processes absent . . . . . . . . . . . . . . . . . . . . . . . . . . . . . . . . . . . **8**

–. Areolation between egg processes present . . . . . . . . . . . . . . . . . . . . . . . . . . . . . . . . . . . **9**

8. Lunules under claws IV dentate, eggs of *beotiae* type . . . . . . . . . . . . . .. . . . . . .. . . . *Pam. beotiae*

–. Lunules under claws IV smooth, egg of *chieregoi* type . . . . . . . . . . . . . . . . . *Pam. chieregoi*

9. Eggs of *submorulatus* type . . . . . . . . . . . . . . . . . . . . . . . . . . . . . . . .. . . . . . . *Pam. submorulatus*

–. Eggs of *richtersi* or *areolatus* type . . . . . . . . . . . . . . . . . . . . . . . . . . . . . . . . . . . . . . . . **10**

10. Eggs of *richtersi* type . . . . . . . . . . . . . . . . . . . . . . . . . . . . . . . . . . . . . . . . . . . . . . . . . **11**

–. Eggs of *areolatus* type . . . . . . . . . . . . . . . . . . . . . . . . . . . . . . . . . . . . . . . . . . . . . . . . .. . **26**

11. Only five or six areoles present around each egg process . . . . . . . . . . . . . . . . . . . . . . . **12**

–. The number of areoles around each egg process larger than six . . . . . . .. . . . . . . . . . **18**

12. Eyes present. . . . . . . . . . . . . . . .. . . . . . . . . . . . . . . . . . . . . . . . . . . . . . . . . . . . . .*Pam. priviterae*

–. Eyes absent. . . . . . . . . . . .. . . . . . . . . . . . . . . . . . . . . . . . . . .. . . . . . . . . . . . . . . . . . . . . . **13**

13. Granulation on leg I–III present. . . . . . . . . . . . . . . . . . . . . . . . . . . . . . . . . . . . .. . . . . . . **14**

–. Granulation on legs I–III absent. . . . . . . . . . . . . . . . . . . . . . . . . . . . . . . . . . . *Pam. depressus*

14. The *pt* values of the macroplacoid length less than *43.5*. . .. . . .. . . . . . . . . .. *Pam. pius*

–. The *pt* values of the macroplacoid length more than *49.0*. . .. . . . . . . . . . . . . . . .. . . **15**

15. Egg process jagged . . . . . . . . . . . . . . . . . . . . . . . . . . . . . .. . . . .. . . . .. . .. . . . . . . . **16**

–. Egg process not jagged . . . . . . . . . . . . . . . . . . . . . . . . . . . . . .. . . . . . . .. . . . . . . . . . . .**17**

16. Egg processes height less than 15.0 µm and parthenogenetic mode of reproduction. . . . . . . . . . . . . . . . . . . . . . . . . . . . . . . . . . . . . . . . . . . . . . . . . . . . . . . . . . . . *Pam. fairbanksi*

–. Egg processes height more than 15.1 µm and bisexual mode of reproduction. . . . . . . . . . . . . . . . . . . . . . . . . . . . . . . . . . . . . . . . . . . . . . . . . . . . . . . . . . . . . . . . . . . . *Pam. celsus*

17. Egg diameter without processes less than 62.5 µm. . . . . . . . . . . . . . . . . . . . . . ***Pam. arduus***

–. Egg diameter without processes more than 65.0 µm. . . . . . . . . . . . . . . . . . . .*Pam. spatialis*

18. Eyes present . . . . . . . . . . . . . . . . . . . . . . . . . . . . . . . . . . . . . . . . . . . . . . . . . . . . . . **19**

–. Eyes absent . . . . . . . . . . . . . . . . . . . . . . . . . . . . . . . . . . . . . . . . . . . . . . . . . . . . . . . **21**

19. Granulation on legs I–III present . . . . . . . . . . . .. . . . . . . . . . . . . . . . . . . . . . . . . . . 20

–. Granulation on legs I–III absent . . . . . . . . . . . . . . . . . . . . . . . . . . . . . . . *Pam. magdalenae*

20. Egg bare diameter less than 87.9 µm, egg process height more than 15 µm, egg processes hemispherical with blunt terminal part . . . . . . . . . . . . . . . . . . . . . . . . . .*Pam. sklodowskae*

–. Egg bare diameter more than 92.0 µm, egg process height less than 13.5 µm, egg processes blunt cones . . . . . . . . . . . . . . . . . . . . . . ***Pam. sagani***

21. Lunules under claws IV dentate. . . . . . . . . .. . . . . . . . . . . . . . . . .. . . . . . . . . ***Pam. alekseevi***

–. Lunules under claws IV smooth. . . . . . . . . .. . . . . . . . . . . . . . . . ..  . . . . . . . . . . . **22**

22. Egg processes with cap-like vesicular structures on the top . . . . . . . . . . . . . . . . . . . **23**

– Egg processes without cap-like vesicular structures on the top. . . . . . . . . . . . . . . . . . . **24**

23. Egg processes with elongated terminal portion, second macroplacoid length less than 6.5 µm, *pt* values of second macroplacoid length less than *14.0*, *pt* values of macroplacoid row length less than *59.0*, placoid row length less than 34.5 µm and *pt* values of placoid row length less than *74.0*. . .. . . . . . . . . . . . . . . . . .. . . . .. . . . .. . . . . . . ***Pam. filipi***

– Egg processes without elongated terminal portion, second macroplacoid length 7.0 µm or more, *pt* values of second macroplacoid length more than *15.0*, *pt* values of macroplacoid row length more than *60.0*, placoid row length more than 34.9 µm and *pt* values of placoid row length more than *77.5*. . . . . .. . . . . ... . . . . . . .. . ... . . . . . . .. ... . . . . . ***Pam. gadabouti***

24. Egg processes with long, thin and flexible terminal portion and egg process height more than 24.5 µm . . . . . . . .. . . . . . .. . . . . .. . . . . . .. . . . . .. . . . . .. . .... ***Pam. lorenae***

– Egg processes without long, thin and flexible terminal portions and egg process height less than 22.5 µm . . . . . . . .. . . . . . .. . . . .. . . . . . .. . . . . . .. . . . . .. . . . . .. . .. . **25**

25. Presence of fine granulation on I–III pair of leg and egg process height more than 15.0 µm . . . . . . . .. . . . . . .. . .. . . . . . .. . . . . . .. . .. . . . . .. . . . . .. . . . . . .. . . . . . ***Pam. richtersi***

–. Absence of fine granulation on I–III pair of leg and egg process height less than 17.0 µm. . . . . . . .. . . . . . .. . . .. . . . . . .. . .. . . . . .. . . . . . .. . . . .. . . . . . .. . . . . . ***Pam. gerlachae***

26. Cuticle with oval pores, egg processes with cap-like structure on the top and clearly narrower under caps . . . . . . . . . . . . . . . . . . . . . . . . . . . . . . . . . . . . . . . . . . .*Pam. garynahi*

–. Cuticle without oval pores, egg processes without cap-like structure on the top and without narrowing at the top . . . . . . . . . . . . . . . . . . . . . . . . . . . . . . . . . . . . . . . . . . . . . **27**

27. Egg processes cone with blunt apex not divided and without elongated terminal part . . . . . . . . . . . . . . . . . . . . . . . . . . . . . . . . . . . . . . . . . . . . . . . . . . . . . . . . . . . . . *Pam. savai*

–. Egg processes different . . . . . . . . . . . . . . . . . . . . . . . . . . . . . . . . . . . . . . . . . . . .. . . . . . . . . . . . **28**

28. Egg processes with long flexible portion on the top i.e. elongated cones . . . . . . . . . . . . . . . . .. . . . *Pam. rioplatensis*

–. Egg processes without long flexible portion on the top . . . . . . . . . . . . . . . . . . . . . . . . . . **29**

29. Egg processes' base width less than 12.5 μm. . . . . . . . . . . . . . . . . . . . . . . . . . *Pam. peteri*

–. Egg processes' base width more than 13.0 μm. . . . . . . . . . . . . . . . . . . . . . . . . . . . . . . . . . **30**

30. Granulation on IV$^{th}$ pair of legs absent and egg processes height more than 15.5 μm . . . . . . . . . . . . . . . . . . . . . . . . . . . . . . . . . . . . . . . . . . . . . . . . . . . . . . . . . . . . . . . . . . . . . . *Pam. hapukuensis*

–. Granulation on IV$^{th}$ pair of legs present and egg processes height less than 15.0 μm . **31**

31. Presence of wrinkled surface on the egg areolae and the absence of cuticular bulge on inner surface of claws I–III. . . . . . . . . . . . . . . . . . . . . . . . . . . . . . . . . . . . . . .*Pam. experimentalis*

–. Lack of wrinkled surface on the egg areolae and the presence of cuticular bulge on inner surface of claws I–III. . . . . . . . . . .. . . . . . . . . . . . . . . . . .. . . . . . . . . . *Pam. metropolitanus*

32. Egg of *csotiensis* type . . . . . . . . . . . . . . . . . . . . . . . . . . . . . . . . . . . . . . . . *Pam. csotiensis*

–. Eggs of *areolatus, huziori, tonollii* or *richtersi* type . . . . . . . . . . . . . . . . . . . . .. . . . . . . . . . . **33**

33. Eggs of *tonollii* type . . . . . . . . . . . . . . . . . . . . . . . . . . . . . . . . . . . . . . . . . . . *Pam. tonollii*

–. Eggs of *areolatus, huziori* or *richtersi* type . . . . . . . . . . . . . . . . . . . . . . . . . . . . . . . .. . . . **34**

34. The egg areolation of the *huziori* type . . . . . . . . . . . . . . . . . . . . . . . . . . . . . . . . . . . . . . **35**

–. Eggs of *richtersi* or *areolatus* type . . . . . . . . . . . . . . . . . . . . . . . . . . . . . . . . . . . . . . . .**36**

35. Only one row of larger teeth are present in the second band in the oral cavity, the distances between all macroplacoids are approximately the same, accessory points are well developed but not protruding high above the primary branch, the diameter of bases of egg processes is approximately equal to or slightly smaller than their height, 9–11 processes are present on the egg circumference . . . . . . . . . . . . . . . . . . . . . . . . . . . . . . . . . . . . . . . . . . . . . . . . . . . . . . . . . . *Pam. huziori*

–. A row of larger teeth and a posterior band of small granules/conical teeth present in the second band of teeth in the oral cavity, the second macroplacoid situated closer to the first than to the third macroplacoid, accessory points extremely well developed, protruding high above the primary branch, diameter of bases of egg processes greater than their height, 12–16 processes on egg circumference . . . . . . . . . . . . . . . . . . . . . . . . . . . .*Pam. derkai*

36. Eggs of *richtersi* type . . . . . . . . . . . . . . . . . . . . . . . . . . . . . . . . . . . . . . . . . . *Pam. spinosus*

–. Eggs of *areolatus* type . . . . . . . . . . . . . . . . . . . . . . . . . . . . . . . . . . . . . . . . . . . . . . . . . **37**

37. The first/anterior band of teeth visible under PCM . . . . . . . . . . . . . . . . . . . . . . . . . . . **38**

–. The first/anterior band of teeth absent or not visible under PCM . . . . . . . . . . . . ***Pam. intii***

38. Lunules under claws IV smooth. . . . . . . . . . . . . . . . . . . . . . . . . . . . . . . . . . . . . . . . . . . . . **39**

–. Lunules under claws IV dentate . . . . . . . . . . . . . . . . . . . . . . . . . . . . . . . . . . . . . . . . . . . .**40**

39. Eyes present, macroplacoid length sequence 2 < 3 < 1, full egg diameter more than 93.0 µm and egg process height more than 17.5 µm. . . . . . . . . . . . . . . . ..  . . . . . . . ***Pam. lachowskae***

–. Eyes absent, macroplacoid length sequence 2 < 1 < 3, full egg diameter less than 92.0 µm and egg process height less than 11.5 µm . . . . . . . . . . . . . . . . . . . . . . . . . . . . . ***Pam. centesimus***

40. Eyes present, macroplacoid length sequence 2 < 1 < 3 and egg processes elongated cone . . . .**41**

–. Eyes absent, macroplacoid length sequence 2 < 3 < 1 and egg processes simple cone. . . . . . . . . . . . . . . . . . . . . . . . . . . . . . . . . . . . . . . . . . . . . . . . . . . . . . . . . . . . . . . . . . . . . . . ***Pam. klymenki***

41. Egg process height more than 26.5 µm and egg process surface smooth . . . . . . . . . . . . . . . . . . . . . . . . . . . . . . . . . . . . . . . . . . . . . . . . . . . . . . . . . . . . . . . . . . . . . . . . . . ***Pam. areolatus***

–. Egg process height less than 17.5 µm and egg process surface apically covered by irregular granulation . . . . . . . . . . . . . . . . . . . . . . . . . . . . . . . . . ..  . . . . . . . . . . . . . . . . . . . . . *** Pam. walteri***

## 11. Conclusions

The genus *Paramacrobiotus* shows a cosmopolitan distribution with the presence of both bisexual and parthenogenetic species. Although the integrative descriptions and redescriptions are improving the overall situation and allowing for new opportunities for detailed study, the phylogeny of the genus *Paramacrobiotus* seems to be unresolved. Also, there are many other studies regarding the life history, cryptobiotic abilities and microbiome community, as well as bacterial endosymbiont infections identification, which are lacking, and such studies are required for the advancement of knowledge of tardigrades in general.

**Supplementary Materials:** The following supporting information can be downloaded at https://www.mdpi.com/article/10.3390/d15090977/s1. SM.01 Locations, mode of reproduction and presence of genetic data for all the *Paramacrobiotus* species. SM.02, Estimates of evolutionary divergence between COI barcode sequences based on p-distances.

**Author Contributions:** Conceptualization, P.K. and Ł.K.; methodology, P.K. and M.M.; formal analysis, P.K. and M.M.; investigation, P.K.; data curation, P.K.; writing—original draft preparation, P.K.; writing—review and editing, P.K., M.M., J.W. and Ł.K.; visualization, P.K. and M.M.; supervision, Ł.K. All authors have read and agreed to the published version of the manuscript.

**Funding:** P.K. is scholarship holder of Passport to the Future—Interdisciplinary doctoral studies at the Faculty of Biology, Adam Mickiewicz University, Poznań POWR.03.02.00-00-I006/17. The work of M.M. was supported by National Science Centre, Poland, grant no. 2021/43/D/NZ8/00344 and grant no. 1220/146/2021 from the Small Grants Pro-gramme of the University of Gdansk (i.e., Ugrants-first competition).

**Data Availability Statement:** All the DNA sequences data are from GenBank.

**Acknowledgments:** Studies have been partially conducted in the framework of activities of BARg (Biodiversity and Astrobiology Research group). We would like to thank Tomasz Bartylak for helping with QGIS.

**Conflicts of Interest:** The authors declare no conflict of interest.

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
