# Peer review of "A Review on the Genus Paramacrobiotus (Tardigrada) with a New Diagnostic Key"

_diversity, doi:10.3390/d15090977_

Round 1

Reviewer 1 Report

Thank you for giving me this opportunity to review the manuscript for summarizing the genus Paramacrobiotus. To date, as you said, taxon of the genus is increased to 45 species and it makes identification for the species difficult and confuse. I actually hope that some review of the genus would be published.

Your effort for this manuscript should be commended, but I think the manuscript is required some revisions for make it better.

General comment;

  1. You must mention about Hara et al., 2021 (doi: https://doi.org/10.1098/rsob.200413) which firstly published genome and transcriptome data of the genus Paramacrobiotus. They have also discussed about desiccation tolerance of a Paramacrobiotus species. I would like to claim to make a new section about Genomic information for the genus. Boothby et al., 2017 has also published a transcriptome data of the genus, so you should mention it too.
  2. The first section “Cryptobiosis” is contrived. The sections of “Morphological” and “Molecular” taxonomy must be placed at first. In addition, you has to explain that what is the difference between Paramacrobiotus and Macrobiotus, which had been involved into single genera.

P1, line 46: Put Pam. peteri in italics.

P1, line 28-: And unify the format of authority (e.g. with () or not).

P2, line 74: Paramacrobiotus tonollii on this line could be abbreviated.

P2, line 89-93: Tsujimoto et al. did not show the study.

P2, line 64-: You have to mention about Pam. metropolitanus which was investigated in Hara et al., 2021.

P4, Fig.1: This figure has some of error. At least, a point of Pam. metropolitanus from Japan is absent, I found. Please recheck them. In addition, the meaning of the color of each point has to be provided in the legend.

P5, line 116: Please add citations to each preys.

P6, line 177: Full name of OTUs is already present. It could be abbreviated in this sentence. Line 187 is also.

P7, line 211-: Which order are the species listed? Unify the order as ABC or old-new or anything. And I found bisexual lineage of Pam. metropolitanus in the list. In addition, I strongly recommend to add karyotype information on this section, because Dr. Rebecchi and her colleagues suggested that the reproductive mode depends on its karyotypes (e.g. diploid, triploid) in Rebecchi et al., 2002, Chromosome Research (doi: 10.1023/A:1020949228862). More information in reproductive apparatus, as well as sperm morphology, must be included in this section. 

P7, line 232: Pam. vanescens has to be italic.

P16, Molecular taxonomy: You has to refer to and mention about Stec et al., 2020, Zool J Linn Soc Lond (doi: https://doi.org/10.1093/zoolinnean/zlz163). They performed cross-strain experiment to observe the molecular taxonomy on the genus.

P17, Fig.2: This Figure is not informative. At first, emphatic reason, why the phylogenetic analysis is limited with using COI sequences, is required. In addition, the tree did not provide newly or originality insight comparing with any other published trees. At least, I would like to claim re-try to analyze with 1. including all taxon which COI sequences are available, 2. check the robustness of the tree by 2 or more analyses (not only BI, but also ML or other). And table 2 should be replaced to supplementary. It is too large on the limited size of the manuscript.

P19, Key for identification: The flow is very nice. I would like to request you to add some figures/illustrations to the key to make understand easier.

Author Response

Dear Editor,

Thank you very much for the constructive reviews and comments on our manuscript. We have addressed all comments made by the Reviewers with our answers (in red) and marked them by ‘REPLY’. We also send you the corrected manuscript with all changes made visible (‘tracked’).

Kind Regards,

Authors

Reviewer 1

Reply: Thank you for the thorough review of our work and expressed criticism.

Reviewer 2 Report

Apart from the recommendation to give more clear definitions for the egg processes types (to give also some schematic drawings will be perfect) I have only some technical notes. You can find them in the attached file.

Author Response

Thank you very much for the constructive reviews and comments on our manuscript. We have addressed all comments made by the Reviewers with our answers (in red) and marked them by ‘REPLY’. We also send you the corrected manuscript with all changes made visible (‘tracked’).

Kind Regards,

Authors

Reviewer 3 Report

COMMENT: I noticed on page 6 (Table 1), last line (Paramacrobiotus vanescens) that the species is reported as with "smooth" cuticle (according to the original description), but Pilato, Binda, Napolitano, & Moncada (2001) amended that by reporting a faint cuticular "punctuation" in the species, and this should be surely taken into account in the Table and the whole
Manuscript. I also want to stress that the old so-called "punctuation" is not always synonym of granulation (I am writing a paper on this). Besides, maybe
it is useful for the Authors to have notice of another amendment on
the description of P. vanescens, by Binda & Pilato 2001 who reported
sculptured areolae (reported as smooth in the original description). 

That's all, looking forward to seeing the manuscript published.

Author Response

(The authors gave the same response as above.)

Round 2

Reviewer 1 Report

Dear authors,

Thank you for your corrections and revisions.

The new figure would make readers well understand, and the descriptions becomes more clearly.

I found some of small mistakes/recommendations on the revised manuscript, so please check my comments again.

Best,

Line 37-: I understood your reply as the ()s should be present to distint as the species from the other genera or not. Please mention it on the manuscript.

Line 127, 169: ITS2 may requires - between S and 2.

Line 155: 1000 should be written as 1,000 as present in line 20 (1,500 species).

Line 283: add a space as: 10 days and " " 10 days, respectively.

Line 357-: add descriptions about spem morphology changes, fertilization, and spem motility mentioned in Sugiura and Matsumoto 2021, Zygote (https://doi.org/10.1017/S0967199420000490) and Sugiura et al. 2022, BMC Zool. (https://bmczool.biomedcentral.com/articles/10.1186/s40850-022-00109-w) etc.

Author Response

(The authors gave the same response as above.)
